# The WRKY28-BRC1 Transcription Factor Module Controls Shoot Branching in *Brassica napus*

**DOI:** 10.3390/plants14030486

**Published:** 2025-02-06

**Authors:** Ka Zhang, Jinfang Zhang, Cheng Cui, Liang Chai, Benchuan Zheng, Liangcai Jiang, Haojie Li

**Affiliations:** Crop Research Institute, Sichuan Academy of Agricultural Sciences, Environment-Friendly Crop Germplasm Innovation and Genetic Improvement Key Laboratory of Sichuan Province, Chengdu 610066, China; zhangka_rapeseed@163.com (K.Z.); zhangjinfang567@163.com (J.Z.); cuicheng005@163.com (C.C.); chailiang1982@126.com (L.C.); zhengbenchuan1987@163.com (B.Z.); jlcrape@163.com (L.J.)

**Keywords:** *Brassica napus*, axillary meristem, branching, WRKY transcription factor, TCP transcription factor

## Abstract

The trade-off between growth and defense is common in plants. We previously demonstrated that BnaA03.WRKY28 weakened resistance strength but promoted shoot branching in *Brassica napus* (rapeseed). However, the molecular mechanism by which WRKY28 promotes branching formation is still obscure. In this study, we found that *BnaA01.BRC1*, *BnaC01.BRC1, and BnaC03.BRC1* are mainly expressed in the leaf axils and contained W-box *cis*-acting elements in the promoter regions. BnaA03.WRKY28 directly bound to the promoter regions of these three copies and inhibited their expression. The *brc1* mutants, the *BnaA01.BRC1*, *BnaC01.BRC1 BnaA03.BRC1* and *BnaC03.BRC1* were simultaneously knocked out, mediated by CRISPR/Cas9, and exhibited excessive branching. The expression level of the ABA biosynthesis encoding gene *NCED3* was significantly reduced in the mutant compared to that in the WT. Instead, the expression level of the ABA catabolism encoding gene *CYP707A3* was significantly higher than that in WT. These results suggest that the excessive branching of the *brc1* mutant may be caused by the release of ABA-mediated bud dormancy. This study provides direct evidence for the potential mechanism of the WRKY28-BRC1 transcription factor module contributing to shoot branching in rapeseed.

## 1. Introduction

*Brassica napus* (rapeseed) is one of the most important oilseed crops worldwide, with a long history of cultivation. The allopolyploid species *B. napus* contains A and C subgenomes derived from the diploid donors *Brassica rapa* and *Brassica oleracea*, respectively [1]. The yield per unit area of rapeseed is determined by the number of siliques, the number of seeds per silique, and the thousand-grain weight, among which the number of siliques contributes the most to the yield [2]. The number of branches is significantly positively correlated with the total number of siliques and is a critical plant architecture trait [3,4].

The shoot branches of the plant develop from axillary meristem (AM). AMs originating in the leaf axils first form dormant buds that either remain dormant until reactivated during subsequent development and further develop into lateral branches or remain silent throughout the life cycle [3,5]. Therefore, the final number of shoot branches is determined not only by the number of AMs but also by the activity of axillary buds. According to the chronological order, the developmental process of the axillary bud can be divided into four stages: axillary bud initiation, dormancy, inhibition release, and sustained development [6]. Transcription factors (TFs), plant hormones, and sugars are involved in AM formation and subsequent bud outgrowth [6,7,8]. *LATERAL SUPPRESSOR* (*LAS*), encoding a member of the GRAS family transcriptional regulator, and *REGULATOR OF AXILLARY MERISTEMS* (*RAX*), encoding a sub-family of R2R3 MYB transcription factors, are expressed in the early stage of axillary bud initiation; the *las* mutants and *rax* mutants are unable to form lateral shoots during vegetative development [9,10]. *CUP-SHAPED COTYLEDON* (*CUC*) is an NAC domain TF encoding gene and a prerequisite for axillary bud initiation. CUC3 plays a primary role in the establishment of boundaries that contain cells with low proliferation and differentiation rates. CUC3 function is partially redundant with that of CUC1 and CUC2 in the establishment of the cotyledon boundary and the shoot meristem. The CUC-mediated establishment of such boundaries may be essential for the initiation of shoot meristems [11]. Whether the formed axillary buds are in a dormant state or activated to outgrowth is regulated by environmental signals mediated by plant hormones, including auxin (IAA), cytokinin, strigolactone (SL), and abscisic acid (ABA). [7,12]. Auxin synthesized in young apical leaves is polarly transported to the leaf axil and inhibits lateral bud outgrowth [13,14]. The auxin efflux carrier PIN-FORMED1 (PIN1) crucially contributes to auxin polar transport in the shoot [15], minimizing leaf axil auxin to promote AM initiation and bud activation [16]. Cytokinins have a direct promoting effect on axillary bud growth. Basally applied cytokinin can release lateral buds from inhibition by apical auxin, which largely depends on AUXIN RESISTANT 1 (AXR1) [17]. SL plays an important role in inhibiting plant branching [18]. The *MORE AXILLARY GROWTH* (*MAX*) gene encodes a protein with similarity to carotenoid cleaving deoxygenase, involved in SL biosynthesis and signal transduction [19]. The *max* mutants exhibit increased shoot branching [20]. ABA is related to seed dormancy, and it is reported that decapitation-induced axillary bud outgrowth is accompanied by a decrease in ABA levels in the buds [21].

Teosinte branched1/Cycloidea/Proliferating cell factor (TCP) is a plant-specific TF family that is involved in plant growth and development [22]. TCP transcription factors contain highly conserved TCP domains. The TCP domain consists of a basic region and a helix loop helix region, which, respectively, act on DNA binding and protein dimerization [23]. ZmTB1, the first identified member of the TCP transcription factors, negatively regulates the growth and development of maize axillary buds [24]. Overexpression of the orthologous gene *OsTB1* in rice can inhibit the growth of lateral buds into branches, thereby reducing tillering [25]. In *Arabidopsis thaliana*, TCP18 (BRC1) is an ortholog of TB1 and a key factor in shoot branching. *AtBRC1* is expressed in developing buds, repressing bud development. *AtBRC1* downregulation leads to branch outgrowth [26]. TCP transcription factors are widely involved in IAA/SL-mediated pathways. AtTCP3 regulates IAA transport by affecting the distribution of PIN proteins, and overexpression of *AtTCP3* causes downregulation of *PIN1* expression [27]. The expression levels of *AtBRC1* are very low in SL-related mutants [26], and it is suggested that BRC1 is a primary target of SL action [28].

The WRKY superfamily TFs, named by the conserved WRKYGQK motif, not only play roles in abiotic and biotic stress response but are also involved in plant growth and development [29,30]. WRKY transcription factors specifically bind to T/CTGACC/T (W-box) *cis*-acting elements of target genes [30]. Overexpression of *AtWRKY71* significantly enhanced the activity of axillary meristem and produced more shoot branches. AtWRKY71 promotes axillary meristem initiation by positively regulating the key branch meristem formation-related TFs, including AtRAX1, AtRAX2, and AtRAX3 [31]. In *Arabidopsis thaliana*, WRKY28 is reported to be involved in the fate determination of megaspore mother cells (MMCs) and the regulation of leaf senescence [32,33]. In the development process that restricts the differentiation of multiple adjacent somatic cells of MMC into MMCs, *AtWRKY28* is specifically expressed in the hypodermal somatic cells surrounding the MMC, inhibiting these somatic cells from gaining the fate of MMC-like fate [32]. FAR-RED ELONGATED HYPOCOTYL3 (FHY3) physically binds to the promoter of *WRKY28* and inhibits the expression of *WRKY28*, thereby negatively regulating leaf senescence [33]. A previous study demonstrated that WRKY28 curbs WRKY33-mediated resistance to *Sclerotinia sclerotiorum* in rapeseed. Under constant infection by *S. sclerotiorum*, BnaA03.WRKY28 interacts with BnaA09.VQ12 and takes precedence over phosphorylated BnWRKY33 to bind to the *BnWRKY33* promoter, thereby weakening resistance but promoting shoot branching [34].

It was preliminarily revealed that BnaA03.WRKY28 induced axillary bud outgrowth and branch formation by targeting *BnaC03.BRC1* promoter. However, the molecular mechanisms involved are still obscure. In this study, we provide direct evidence that three copies of *BRC1* are physically bound by WRKY28, and their expression levels were downregulated, resulting in an increase in axillary buds and lateral branches in rapeseed.

## 2. Results

### 2.1. Five BRC1 Copies Are Identified in the Rapeseed Genome

In order to further explore the role of BRC1 as a downstream of WRKY28 in rapeseed, we first analyzed the sequence of BRC1. Five paralogs of AtBRC1 were identified in the *B. napus* genome, located on chromosomes A01, A03, C01, C03, and C05, and named BnaA01.BRC1, BnaA03.BRC1, BnaC01.BRC1, BnaC03.BRC1, and BnaC05.BRC1, respectively. There were two paralogs of AtBRC1 in the *B. rapa* genome, named BraA01.BRC1 and BraA03.BRC1; however, there were three paralogs of AtBRC1 in the *B. oleracea* genome, named BolC01.BRC1, BolC03.BRC1, and BolC05.BRC1 (Figure 1A). About 7500 years ago, *B. rapa* and *B. oleracea* naturally hybridized to form *B. napus*, which inherits the A genome (A01–A10) and C genome (C01–C09) of *B. rapa* and *B. oleracea*, respectively [1]. Phylogenetic analyses revealed the evolutionary relationship of BRC1 in Brassica species. BnaA01.BRC1 and BraA01.BRC1 are on the same clade, BnaA03.BRC1 and BraA03.BRC1 are on the same clade, BnaC01.BRC1 and BolC01.BRC1 are on the same clade, BnaC03.BRC1 and BolC01.BRC3 are on the same clade, and BnaC05.BRC1 and BolC05.BRC1 are on the same clade (Figure 1A). These results indicate that the evolution of BRC1 follows the rule of genetic inheritance between subgenomes but exhibits asymmetric fragment duplication between subgenomes A and C. Sequence alignment showed that the five *Brassica* BRC1 copies, like AtBRC1, contained conserved TCP domains, but had poor conservation outside of the TCP domains (Figure 1B). Furthermore, the MEME website was used to predict the conserved motifs of these BRC1 proteins. BnaA03.BRC1, BnaC03.BRC1 and BnaC05.BRC1, like AtBRC1, all contained Motif 1–10 and had a quite consistent arrangement. The Motif composition and distribution of BnaA01.BRC1 and BnaC01.BRC1 were similar but differed significantly from AtBRC1. BnaA01.BRC1 and BnaC01.BRC1 only consisted of six motifs (Motif 1, 2, 3, 5, 6, and 8), without Motif 4 at the N-terminus, Motif 9 at the C-terminus, Motif 7 and 10 in the middle (Figure 1C). BnaA01.BRC1 and BnaC01.BRC1 shared 97.18% amino acid sequence identity; BnaA03.BRC1 and BnaC03.BRC1 shared 92.42% amino acid sequence identity. However, the identity between other pairs of BRC1 ranges from 68.26% to 75.61% (Figure 1D).

### 2.2. Multiple Copies of BRC1 May Function Redundantly in Shoot Branching in Rapeseed

In our previous study, it was found that *BnaA03.WRKY28* was expressed in leaf axils at the branching formation stage and *BnaA03.WRKY28 overexpression* lines exhibited high axillary bud activity and excessive branching. BnaA03.WRKY28 is physically bound to the promoter of *BnaC03.BRC1* [34]. To elucidate the biological function of BnaC03.BRC1 in the branching development of rapeseed, we generated transgenic lines with *BnaC03.BRC1* knockout mediated by CRISPR/Cas9. Considering that BnaC03.BRC1 shared more than 90% amino acid identity with BnaA03.BRC1 (Figure 1D), the two paralogous copies were probably functionally redundant. Therefore, the designed sgRNAs targeted both BnaA03.BRC1 and BnaC03.BRC1. Both sgRNA1 and sgRNA2 were located in the first exon, and sgRNA1 is at the forefront of the conserved and functionally important TCP domain (Appendix A). Two homozygous editing lines, *brc1-a3c3*#16 and *brc1-a3c3*#18, were used for subsequent research. In the *brc1-a3c3*#16 line, a base A was inserted at the sgRNA1 targeting sites on chromosomes A03 and C03, while a base A was deleted at the sgRNA2 targeting sites on chromosomes A03 and C03. In the *brc1-a3c3*#18 line, on chromosomes A03 and C03, a base G and a base A were inserted at the sgRNA1 and sgRNA2, respectively (Appendix A). No obvious differences were observed in axillary bud development or branching formation in wild-type (WT) plants (Appendix A), *brc1-a3c3*#16 (Appendix A) and *brc1-a3c3*#18 (Appendix A). Real-time quantitative PCR (RT-qPCR) was conducted to detect the expression levels of the five *BRC1* copies in the axillary buds, leaves, and flower buds of rapeseed (Figure 2A, Appendix A). In axillary buds, the expression levels of *BnaA01.BRC1*, *BnaC01.BRC1, and BnaC03.BRC1* were significantly higher than the other two copies (Figure 2A). Moreover, the transcription abundance of *BnaA01.BRC1*, *BnaC01.BRC1, and BnaC03.BRC1* mainly accumulated in the leaf axils, with less accumulation in the leaves and flower buds (Appendix A). In *BnaA03.WRKY28 overexpression* lines, the expression levels of *BnaA01.BRC1*, *BnaC01.BRC1, and BnaC03.BRC1* were significantly lower than those in the wild type (Figure 2B,C,E), while there was no difference in the expression levels of *BnaA03.BRC1* and *BnaC05.BRC1* between the *BnaA03.WRKY28 overexpression* lines and the wild type (Figure 2D,F). These results indicate that the three copies of BnaA01.BRC1, BnaC01.BRC1, and BnaC03.BRC1 may have functional redundancy in the axillary bud development of rapeseed.

### 2.3. BnaA03.WRKY28 Targets BnaA01.BRC1, BnaC01.BRC1, and BnaC03.BRC1 and Negatively Regulates Their Expression

It is well documented that the WRKY TF regulates the expression of downstream genes by binding to the W-box *cis*-element with the core sequence T/CTGACC/T in the target gene promoter [30]. We analyzed the 2000 bp sequences upstream of start codon ATG of the five rapeseed *BRC1* copies. The *BnaA01.BRC1* and *BnaC01.BRC1* copies were sequence conserved, both containing a typical W-box at approximately -260 bp. The *BnaC03.BRC1* copy contained two typical W-boxes at approximately -1040 bp and -1115 bp. However, no W-boxes were found in the *BnaA03.BRC1* and *BnaC05.BRC1* copies (Figure 3A). We demonstrated that BnaA03.WRKY28 bound to the *BnaC03.BRC1* promoter via the above-mentioned two W-boxes [34]. We expressed the BnaA03.WRKY28 protein through *Escherichia coli* and synthesized the probe containing W-box from *BnaA01.BRC1*/*BnaC01.BRC1* promoter region for carrying out electrophoretic mobility shift assay (EMSA). The mutated probe (replacing a segment of TTGACC with AGAGAG) was synthesized as control. When the BnaA03.WRKY28 protein was added, the migration of the fluorescein-labeled probe was shifted dramatically, but not the mutated control probe (Figure 3B). This shifted band was completely weakened when the unlabeled probe was added (Figure 3B). These results indicate that BnaA03.WRKY28 directly binds to the promoter region of *BnaA01.BRC1*/*BnaC01.BRC1*.

To test the effect of BnaA03.WRKY28 binding to the *BnaA01.BRC1*/*BnaC01.BRC1/BnaC03.BRC1* promoter on the transcription of *BnaA01.BRC1*/*BnaC01.BRC1/BnaC03.BRC1*, we generated the effector (WRKY28) in which BnaA03.WRKY28 expression was driven by a CaMV 35S promoter and the reporters (promoter A01, promoter C01, and promoter C03) in which the promoter regions including W-boxes were fused to the firefly luciferase gene. The effector or control (vector without BnaA03.WRKY28) was cotransformed into tobacco leaves with promoter A01, promoter C01, or promoter C03, respectively. The fluorescence was weaker in the WRKY28 cotransformed with promoter A01 (Figure 4A,B,G), promoter C01 (Figure 4C,D,G), or promoter C03 (Figure 4E,F,G) than in controls, suggesting that BnaA03.WRKY28 could recognize the *BnaA01.BRC1*/*BnaC01.BRC1/BnaC03.BRC1* promoter and inhibited its expression.

BnaA03.WRKY28 targets multiple BRC1 copies and negatively regulates their expression. (A) Characterization of the promoter sequences of three BRC1 copies of rapeseed and the positions of W-box. The core sequence of the W-box is marked with red. (B) EMSA to test the interaction between BnaA03.WRKY28 and the promoters of *BnaA01.BRC1* and *BnaC01.BRC1* in vitro. The sequences of *BnaA01.BRC1* and *BnaC01.BRC1* promoters used as probes are the same. Arrow indicates the position of shifted bands in each lane. Probes labeled with Cy5 were used. The mutated probe replaced the core W-box sequence with AGAGAG; His-WRKY28, BnaA03.WRKY28 protein fused to his tag. (C~I) Dual-luciferase transient transcriptional activity assays for detecting the relative luciferase activity (the ratio of firefly luciferase activity to *Renilla* luciferase activity) when cotransformed with effector and reporter. Fluorescence scanning in *Nicotiana benthamiana* leaves with *BnaA01.BRC1* promoter (C), *BnaC01.BRC1* promoter (E) and *BnaC03.BRC1* promoter (G) as a reporter. (D,F,H,I) Quantifying and statistically analyzing the fluorescence values in cotransformed *N.benthamiana* leaves. Data for D, F, and H were collected from the fluorescence values from the images of C, E, and G by InDiGo software, respectively. (I) Data were measured using luminometer after enzymatic reactions according to the instructions of the Promega manufacturer. Data are shown as means ± SD (*n* = 3). Asterisks indicate significant differences compared with WT (*t*-test; * *p* < 0.05).

### 2.4. The brc1 Mutants Exhibit Excessive Axillary Branching Formation

We designed sgRNA3 and sgRNA4 that could simultaneously target five *BRC1* copies of rapeseed, and both guide RNAs were located within the TCP domain (Figure 5A). We obtained some CRISPR/Cas9-mediated homozygous editing lines, among which *brc1-a1c1a3c3c5*#4 and *brc1-a1c1a3c5*#21 (*brc1* mutants) were used for subsequent research. In *brc1-a1c1a3c3c5*#4, a 34 bp fragment located between sgRNA3 and sgRNA4 was deleted on chromosomes A01, C01, A03, and C03, while no base edited on chromosome C05 (Figure 5B). In *brc1-a1c1a3c5*#21, the 34 bp fragment above mentioned was also deleted on chromosomes A01, C01, and C03, while on chromosomes A03 and C05, the 5 bp GACAT was deleted at the position sgRNA3 targeted (Figure 5B). During the vegetative stage when the plants were approximately six weeks old, we found significant differences in the number of axillary buds and branches between the *brc1* mutants and WT. In WT, there was only one growth point on the apical meristem of the main stem (Figure 5C,D). However, in the *brc1* mutants, axillary branches elongated from the main stem below the apical growth point. Each extra axillary branch had a growth point with normal growth and development ability, around which new leaves grow (Figure 5C,E,F). To confirm this phenotypic difference, we observed the shoot apical meristem of approximately four-week-old seedlings in paraffin sections. It was found that the axillary buds of the *brc1* mutants had obvious outgrowth compared to WT (Appendix A). As the plants grew older, we conducted statistical analysis on the branches and growth points of WT, *brc1* mutants, and *brc1-a3c3* lines. The WT plants, as well as *brc1-a3c3* lines, had only one growth point without an axillary branch, while the *brc1* mutants had 3–5 growth points (Figure 5G,H), which subsequently formed excessive effective branching (Figure 5I). The significant increase in growth points in the *brc1* mutants was due to the outgrowth of axillary buds below the growth point of the main stem. These results suggest that simultaneous mutation of BnaA01.BRC1, BnaC01.BRC1, BnaA03.BRC1, and BnaC03.BRC1 promotes axillary bud activity and shoot branching in rapeseed (Figure 5J).

### 2.5. BRC1 May Regulate Axillary Bud Dormancy and Outgrowth Through Hormone Signaling in Rapeseed

The formation of AMs at leaf axils and the activity of formed axillary buds mainly determine the number of plant branches [7]. TFs mediated gene expression regulation and plant hormones mediated signal transduction play important roles in the above two growth and development processes [6,7,8]. It is reported that transcription factors LAS, RAX, and CUC are prerequisites for axillary bud initiation and formation [9,10,11]. In addition, IAA, cytokinin, SL, and ABA affect bud initiation and outgrowth [7,12]. To reveal the pathways involved in the influence of BRC1 on branching formation in rapeseed, RT-qPCR was performed for WT, *brc1* mutants, and *brc1-a3c3* lines. *LAS*, *RAX1,* and *CUC3*, which control the early steps of AM formation, showed no significant difference in expression levels among the three genotypes of plants (Figure 6A,B,C). The transcriptional abundance of *PIN1* (acting on auxin polarly transport) in the *brc1* mutants was significantly higher than that in the WT (Figure 6D), but the difference was not significant compared to the *brc1-a3c3* lines (Appendix A); the expression level of *PIN1* showed no significant difference between the *brc1-a3c3* lines and WT (Figure 6D). The transcriptional abundance of *AXR1* (encoding a cytokinin signaling factor involved in AM formation inhibition mediated by auxin) in the *brc1* mutants was significantly lower than that in the WT (Figure 6E), but the difference was not significant compared to the *brc1-a3c3* lines (Appendix A); the expression level of *AXR1* showed no significant difference between the *brc1-a3c3* lines and WT (Figure 6E). MAX1 and MAX2 were related to SL biosynthesis and SL signal transduction, and their expression levels showed no significant difference among the three genotypes of plants (Figure 6F,G). NCED3 is the critical enzyme at the rate-limiting step in ABA biosynthesis [35,36]. The expression level of *NCED3* in the *brc1* mutants was significantly lower than that in WT and the *brc1-a3c3* lines, while the difference was not significant in the WT and the *brc1-a3c3* lines (Figure 6H and Appendix A). Cytochrome P450 CYP707A encodes ABA 8′–hydroxylase, and CYP707A3 is a key enzyme in ABA catabolism [36,37]. The expression level of *CYP707A3* in the *brc1* mutants was significantly higher than that in WT and the *brc1-a3c3* lines, while the difference was not significant in the WT and the *brc1-a3c3* lines (Figure 6I and Appendix A). These results suggest that multiple copies of BRC1 play redundant roles in shoot branching in rapeseed, which may be achieved by affecting the accumulation of ABA in the leaf axils, and then acting on the dormancy or release of axillary buds.

## 3. Discussion

The *Brassica* species, including *B. rapa* and *B. oleracea*, have undergone genome duplications in the evolution process [38]. The hybridization of *B. rapa* and *B. oleracea* resulted in the formation of *B. napus* [1]. In general, one gene in *A. arabidopsis* has more than one homolog in the *B. napus* genome. Due to high sequence identity, functional redundancy between paralogous copies is common. BnaA08.PPT1 and BnaC08.PTT1 functions redundantly in plant chloroplast development; and when the PPT1 copies on chromosomes A08 and C08 are knocked out simultaneously, the mutants grow slower with yellowish leaves [39]. BnaA09.TT8 and BnaC09.TT8b play redundant roles in the formation of yellow seed coats in rapeseed seeds [40]. However, some duplicated genes are genetically confirmed to be subfunctionalized. The gene pairs *BnaA09.ZEP/BnaC09.ZEP* and *BnaA07.ZEP/BnaC07.ZEP* encode tissue-specific enzymes involved in carotenoid and ABA biosynthesis in flowers and leaves, respectively, indicating functional divergence [41]. It was identified that there were five paralogs of *AtBRC1* in the rapeseed genome. Simultaneously knocking out four copies resulted in the *brc1* mutant exhibiting excessive branching phenotype. But when only A03 and C03 copies were knocked out simultaneously, no visible differences were observed in branching development, suggesting that multiple BRC1 copies of rapeseed may play redundant roles during branching development. The *brc1* mutants showed excessive branching without *BnaC05.BRC1* edited, and *BnaC05.BRC1* had low transcript abundance at the leaf axil. Compared to other copies, *BnaC05.BRC1* may not be critical for branching development. Gene editing mediated by CRISPR/CRISPR-associated (Cas) systems is widely used to create precisely mutated varieties [42]. This study provided effective sgRNAs for creating increased axillary branch variety in rapeseed. In *A. thaliana*, BRC2 plays a slightly weaker role than BRC1 in shoot branching [26,43]. A total of 80 *BnTCP* genes are identified in the rapeseed genome [44]. Among them, we identified BnaA02.BRC2, BnaC02.BRC2 and BnaC06.BRC2 inherited from the A02 chromosome of *B. rapa*, C02, and C06 chromosome of *B. oleracea*, respectively (Appendix A). BRC1 and BRC2 of *Brassica* species were conserved in the TCP domain and were diverse in other domains (Appendix A). It was found that only motifs 1, 3, and 9 were conserved in multiple BRC1 and BRC2 copies in rapeseed (Appendix A), suggesting functional differentiation.

Intricate pathways are proposed to elucidate the involvement of BRC1 in the development of plant axillary buds. IAA biosynthesized at the top of the main stem is transported downwards to form a polar auxin transport flow [13], promoting the expression of the genes *CCD7* and *CCD8* involved in the biosynthesis of SL [45]. The expression of *BRC1* is promoted when the synthesized SL is transported upwards to axillary buds [19]. BRC1 protein enhances the expression of *HB21*, *HB40,* and *HB53*, thereby promoting the expression of ABA biosynthesis gene *NCED3* and keeping axillary buds dormant [46,47]. ABA accumulation in leaf axils inhibits axillary bud overgrowth [48]. In addition, IAA inhibits the expression of the cytokinin biosynthesis gene *IPT* through AXR1-mediated signal transduction [49,50]. That reduced biosynthesis of CK weakens the inhibition of *BRC1* expression by CK, maintaining bud dormancy [28]. In this study, no significant differences were found in the expression levels of genes related to AM formation and axillary bud initiation between the *brc1* mutant and WT; however, the expression levels of key genes involved in ABA biosynthesis were significantly reduced in the *brc1* mutant, while the expression levels of genes related to ABA metabolism were significantly increased. The excessive branching of the *brc1* mutant may be caused by the release of ABA-mediated bud dormancy.

It is reported that plants produce more basal branches when invaded by pathogens [51]. BnaA03.WRKY28 acts as a braking factor for defense in rapeseed by reducing resistance strength while promoting branching [34]. In this study, it is demonstrated that BnaA03.WRKY28 directly targets multiple copies of *BRC1*, inhibits their expression, and reduces the accumulation of ABA at leaf axils, leading to the release of axillary bud dormancy and axillary bud overgrowth. The WRKY28-BRC1 transcription factor module contributes to the trade-off between defense and growth and may increase seed reproduction and provide the possibility for the plants to ensure their survival in rapeseed when facing deadly diseases.

## 4. Materials and Methods

### 4.1. Plant Materials and Growth Conditions

The rapeseed variety Westar was used as the recipient of gene editing vectors mediated by CRISPR/Cas9. *BnaA03.WRKY28 overexpression* lines were offsprings of the lines used in previous study [34]. Wild-type plants were grown at an experimental field in Chengdu under natural conditions for seed propagation. The gene-edited lines and WT plants were grown in a greenhouse at 22 °C under a long-day condition (16 h day/8 h night). *Nicotiana benthamiana* plants were grown in soil under the same conditions in the greenhouse for transient expression assay and the luciferase assay.

### 4.2. Sequence Alignment and Phylogenetic Analysis

The sequences of *AtBRC1* and *AtBRC1* were obtained from The Arabidopsis Information Resource (TAIR, https://www.arabidopsis.org/, accessed on 10 December 2024) database and submitted to the Brassica napus multi-omics information resource (BnIR, http://yanglab.hzau.edu.cn/BnIR, accessed on 10 December 2024) website to determine the duplications on the rapeseed genome. Sequence alignments were performed using ClustalW in MEGA(MEGA_11.0.10) software and display the results using GENEDOC (GENEDOC 2.7.0.0) software. The rootless phylogenetic trees were constructed using the Maximum Likelihood method with 1000 replicates for bootstrap analysis, and MEGA software (version number, 5.1.3.308) was used to display the result.

### 4.3. Generation of Binary Constructs and Knockout Lines

The CRISPR/Cas9-mediated gene editing constructs were designed based on the binary vector pKSE401 [52]. The sgRNAs were designed using CRISPR-P (http://cbi.hzau.edu.cn/cgi-bin/CRISPR, accessed on 10 December 2024). The constructed plasmids were confirmed by restriction digestion analysis and sequencing before being transformed into *Agrobacterium tumefaciens* GV3101 by electroporation method and transformed into Wild-type Westar using the modified method according to previous description [53]. The mutation sites and variations of transgenic lines were detected by sequencing and visualized through the HI-TOM platform [54]. sgRNAs sequences and specific primers for mutation type determination are listed in Appendix A.

### 4.4. Isolation of RNA and RT-qPCR

Total RNA was extracted from axillary buds, leaves, and flower buds of WT plants or transgenic lines using the RNAprep pure plant kit (TIANGEN, DP441, Beijing, China). Two micrograms of total RNA was used for cDNA synthesis using the RevertAid First Strand cDNA kit (Fermentas, #K1622, Waltham, MA, USA). RT-qPCR analysis was performed with the CFX96^TM^ Real-Time system (Bio-Rad, Hercules, CA, USA) using the SYBR Green Realtime PCR Master Mix (TOYOBO, QPK-201, Tokyo, Japan). BnACTIN2 was used as a control and CT values of WTs were used to normalize expression levels according to the 2^−∆∆CT^ method [55]. The RT-qPCR primers are given in Appendix A; raw CT values and *t*-test results are given in Appendix A. RT-qPCR was performed in three biological replicates.

### 4.5. Electrophoretic Mobility Shift Assay (EMSA)

Two complementary oligonucleotide strands were isolated from the promoter of *BnaA01.BRC1*/*BnaC01.BRC1*, labeled with Cy5, and annealed to generate probes. The recombinant His-WRKY28 (BnaA03.WRKY28 protein fused to his tag) protein was expressed in *Escherichia coli* BL21 with 0.2 mM isopropyl-beta-D-thiogalactopyranoside (IPTG) at 16 °C for 16 h. The cells were lysed using ultrasonic disruptor (SONICS VCX130, USA), and the supernatant was added onto a column equipped with Ni^2 +^ affinity resin (BBI, C600033, China) after centrifugation at 12,000× *g* at 4C for 30 min. The DNA–protein binding activities were incubated in the reaction system containing EMSA/Gel-shift Binding Buffer (Beyotime, Haimen, China, GS005) and 25 nM Cy5-labeled probe at 23 °C for 30 min. For the competition, unlabeled probes were added to the reactions. The reaction mixture was loaded on a 6% native polyacrylamide gel and electrophoresed in 0.5 TBE (45 mM Tris-base, 45 mM boric acid, 0.5 mM EDTA, pH 8.3) at 4 °C for 1 h at 80 V in the dark. Fluorescence-labeled DNA on the gel was directly detected using Fujifilm FLA-9000 (FujiFilm, Tokyo, Japan). Probe and primer sequences are given in Appendix A. Three biological replicates were conducted for EMSA.

### 4.6. Dual-LUC Transient Transcriptional Activity Assay

BnaA03.WRKY28 was cloned into pGreenII 62-SK vector as effector. The promoter sequence of *BnaA01.BRC1*/*BnaC01.BRC1* or *BnaC03.BRC1* was inserted into the pGreenII 0800-LUC vector to construct reporter The CaMV35S-driven Renilla LUC was used as an internal control. The effector, reporter, and internal control were co-transformed into *Nicotiana benthamiana* leaves. The fluorescent regions on tobacco leaves indicate the LUC signal. The tobacco leaves were imaged with Nightshade LB985 (Germany), and LUC luminescence intensity was analyzed using IndiGO software (version number, 2.0.2.0). LUC activity was measured according to the manufacturer’s instructions (Promega, E1910, Madison, WI, USA). Construct primers are listed in Appendix A. Three biological replicates were conducted for luciferase assays.

### 4.7. Histological Analysis

For histological analysis of shoot apical meristem in the vegetative development stage, the shoot apices of WT plant and the *brc1* mutants at 4-week-old after emergence were fixed with FAA (50% ethanol, 3.7% formaldehyde, and 5% acetic acid), dehydrated in an ethanol series (30%, 50%, 70%, 85%, 95%, and 100%). Xylene was used for transparency before adding crushed paraffin wax. The thickness of paraffin sections was generally 8–10 μm. The sections were stained with 2% toluidine blue solution and imaged using a Nikon DS-Ri 1 microscope.

## Figures and Tables

**Figure 1 plants-14-00486-f001:**
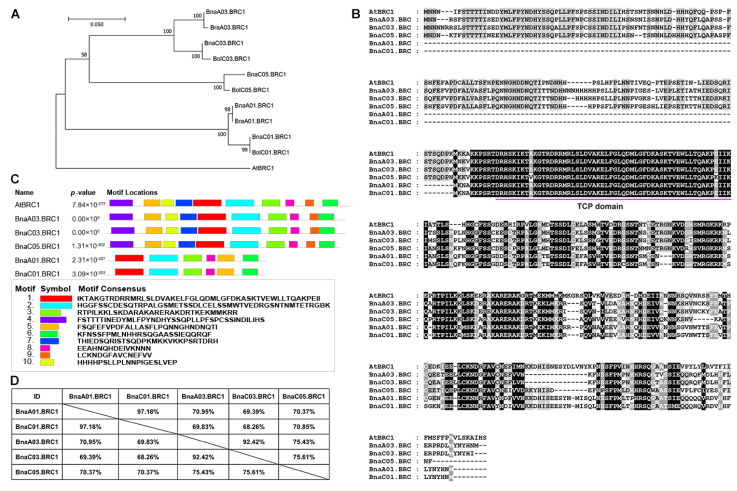
Sequence alignment and conserved domain analysis of BRC1 in rapeseed. (**A**) Phylogenetic tree was constructed to show the evolutionary relationships between each copy of BRC1 in *Brassica* species and *Arabidopsis thaliana*. (**B**) Amino acid sequences alignment of identified BRC1 copies in rapeseed and *Arabidopsis.* The conserved amino acid residues were shown in black shading, and similar residues were displayed in gray shading. The TCP domain was marked with magenta horizontal line. (**C**) Conserved motifs of five BRC1 copies were predicted by MEME. (**D**) Sequence identity of five BRC1 copies in rapeseed based on amino acid sequence. Bna, *Brassica napus*; Bra, *Brassica rapa*; Bol, *Brassica oleracea;* At, *Arabidopsis thaliana*.

**Figure 2 plants-14-00486-f002:**
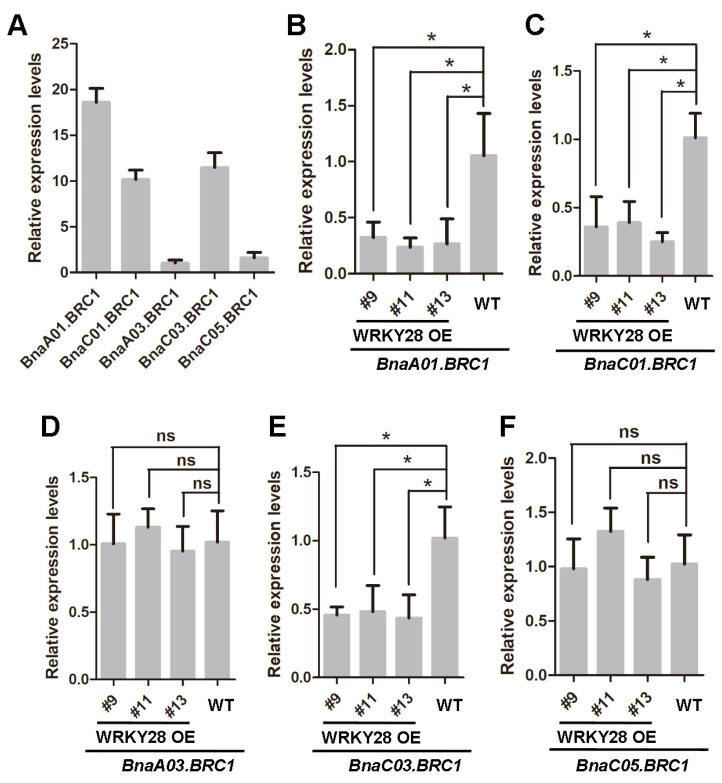
Expression analysis of five *BRC1* copies of rapeseed in AMs of WT and WRKY28 OE lines. (**A**) Expression levels of *BnaA01.BRC1*, *BnaC01.BRC1 BnaA03.BRC1*, *BnaC03.BRC1* and *BnaC05.BRC1* in AMs of wild-type Westar. The mean values of *BnaA03.BRC1* were used to normalize the expression levels. (**B**–**F**) Comparison of transcript abundance of *BnaA01.BRC1* (**B**), *BnaC01.BRC1* (**C**) *BnaA03.BRC1* (**D**), *BnaC03.BRC1* (**E**), and *BnaC05.BRC1* (**F**) between WRKY28 OE lines and WT. #9, #11 and #13 are three representative *BnaA03.WRKY28 overexpression* (WRKY28 OE) lines. Data are shown as means ± SD (*n* = 3). Asterisks indicate significant differences compared with WT (*t*-test; ns, *p* > 0.05, * *p* < 0.05).

**Figure 3 plants-14-00486-f003:**
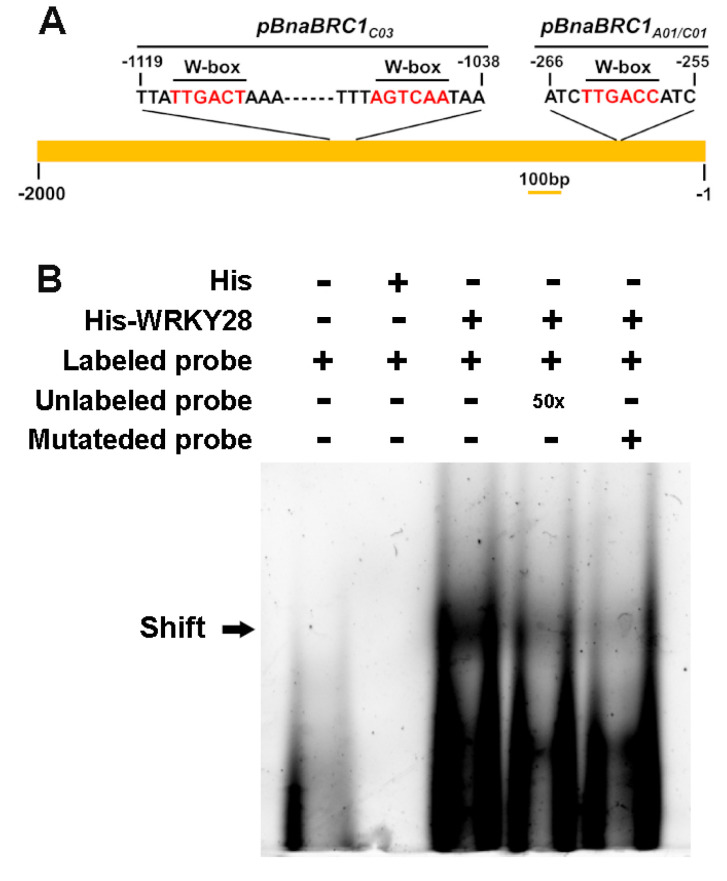
BnaA03.WRKY28 physically targets the promoter of *BnaA01.BRC1*/*BnaC01.BRC1* in vitro. (**A**) Characterization of the promoter sequences of three BRC1 copies of rapeseed and the positions of W-box. The core sequence of the W-box is marked with red. (**B**) EMSA to test the interaction between BnaA03.WRKY28 and the promoters of *BnaA01.BRC1* and *BnaC01.BRC1* in vitro. The sequences of *BnaA01.BRC1* and *BnaC01.BRC1* promoters used as probes are the same. Arrow indicates the position of shifted bands in each lane. Probes labeled with Cy5 were used. The mutated probe replaced the core W-box sequence with AGAGAG; His-WRKY28, BnaA03.WRKY28 protein fused to his tag.

**Figure 4 plants-14-00486-f004:**
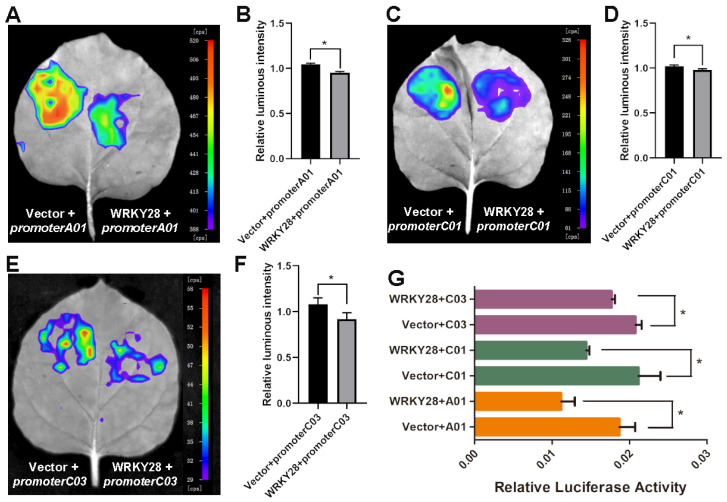
BnaA03.WRKY28 targets multiple BRC1 copies and negatively regulates their expression. Dual-luciferase transient transcriptional activity assays for detecting the relative luciferase activity (the ratio of firefly luciferase activity to *Renilla* luciferase activity) when cotransformed with effector and reporter. Fluorescence scanning in *Nicotiana benthamiana* leaves with *BnaA01.BRC1* promoter (**A**,**B**), *BnaC01.BRC1* promoter (**C**,**D**) and *BnaC03.BRC1* promoter (**E**,**F**) as a reporter. (**B**,**D**,**F**,**G**) Quantifying and statistically analyzing the fluorescence values in cotransformed *N.benthamiana* leaves. Data for (**B**,**D**,**F**) were collected from the fluorescence values from the images of (**A**,**C**,**E**) by InDiGo software (version number, 2.0.2.0), respectively. (G) Data were measured using luminometer after enzymatic reactions according to the instructions of the Promega manufacturer. Data are shown as means ± SD (*n* = 3). Asterisks indicate significant differences compared with WT (*t*-test; * *p* < 0.05).

**Figure 5 plants-14-00486-f005:**
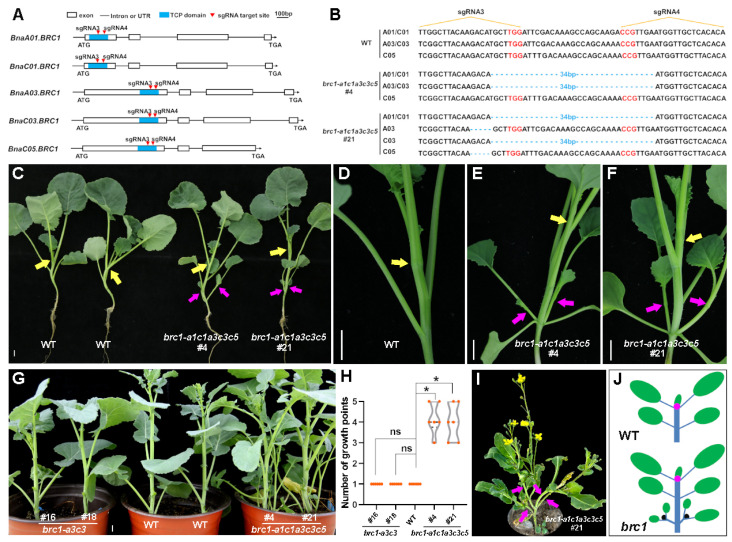
Knockout of multiple copies of *BRC1* leading to excessive branching in rapeseed. (**A**) Schematic diagram to show the sequence characterization of five copies of *BRC1* of rapeseed and the target sites of sgRNAs using the CRISPR/Cas9 system. ATG, start codon; TGA, stop codon. (**B**) The edit types of two independent homozygous mutants are shown. The protospacer-adjacent motifs (PAM) are marked in red. (**C**) Phenotype of branching or buds (indicated by arrows, including shoot apical meristems) in WT (**left**) and the *brc1* mutants (**right**); ca. six-week-old plants were imaged; scale bar, 1 cm. (**D**–**F**) Close-up view of the representative bud regions of WT and the *brc1* mutants in C; scale bar, 1 cm. In (**C**–**F**), main stem buds are indicated by yellow arrows, and axillary buds are indicated by magenta arrows. (**G**,**H**) Statistical analysis of axillary buds in WT, the *brc1* mutants and *brc1-a3c3* lines; the *brc1* mutants, knockout of *BnaA01.BRC1*, *BnaC01.BRC1*, *BnaA03.BRC1* and *BnaC03.BRC1* lines; ca. eight-week-old plants were imaged; scale bar, 5 cm. Data are shown as means ± SD (*n* = 3). Asterisks indicate significant differences compared with WT (*t*-test; ns, *p* > 0.05, * *p* < 0.05). (**I**) Flowering stage of the *brc1-a1c1a3c3c5*#21 line of C; branches are indicated by arrows. (**J**) Schematic of buds in WT and the *brc1* mutants; magenta circles, buds, or growth points of shoot apical meristem; black circles, bud or growth point of axillary meristem meristem; green ellipses, leaves.

**Figure 6 plants-14-00486-f006:**
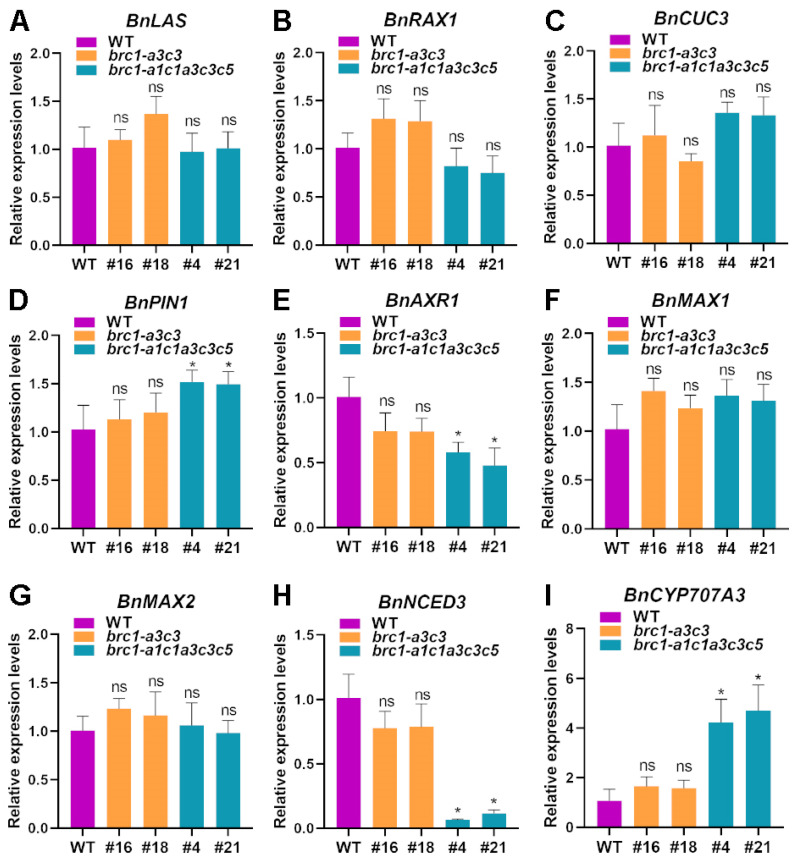
Expression analysis of genes related to branching in WT, the *brc1* mutants, and *brc1-a3c3* lines. (**A**–**C**) Genes involved in AM initiation. (**D**) Gene involved in auxin polarly transport. (**E**) Gene involved in cytokinin signal transduction. (**F**,**G**) Genes related to SL biosynthesis and signal transduction. (**H**) Gene involved in ABA biosynthesis. (**I**) Gene involved in ABA catabolism. Data are shown as means ± SD (*n* = 3). Asterisks indicate significant differences compared with WT (*t*-test; ns, *p* > 0.05, * *p* < 0.05).

## Data Availability

The data that supports the findings of this study are available in the Appendix A of this article.

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
