# Peer review of "The WRKY28-BRC1 Transcription Factor Module Controls Shoot Branching in Brassica napus"

_plants, 2025, doi:10.3390/plants14030486_

Round 1
Reviewer 1 Report
Comments and Suggestions for Authors
The manuscript investigates the molecular mechanism underlying shoot branching in B. napus by exploring the role of the WRKY28-BRC1 transcription factor module. This study provides significant insights into the genetic regulation of branching, contributing to our understanding of plant architecture in oilseed crops. The integration of CRISPR/Cas9 gene editing, gene expression analysis, and promoter-binding assays ensures robust experimental support. The findings are potentially impactful for crop breeding strategies aimed at optimizing plant architecture. In conclusion, the paper would be sufficient to merit publication in Plants, though a revision is recommended, which needs to include the following points.
(1) Many figures in the manuscript are too small, making it difficult to discern details. The following improvements are strongly recommended:
Figure 1
- Increase the font size of labels and axes for better readability, especially in the phylogenetic tree and sequence alignment sections.
- For the motif analysis (Figure 1C), provide an enlarged version to improve the visibility of the sequence motifs.
Figure 3
- In Figure 3A, enlarge the schematic representation of promoter sequences and adjust the text and annotations for better readability.
- The EMSA band shift diagram (Figure 3B) should be enlarged, with clearer labels and a more detailed legend.
- Enhance the resolution of fluorescence images in Figures 3C–H and ensure scale bars are clearly visible.
Figure 4
- In Figure 4C-F, use different colors for the arrows to clearly distinguish between main stem buds and axillary buds.
(2) Specify the number of biological replicates used in EMSA, luciferase assays, and RT-qPCR to enhance reproducibility.
(3) Figure 3A and B: The figure legend states that two types of promoters, BnaA01.BRC1 and BnaC01.BRC1, were analyzed. However, the figure seems to show results for only one. Which promoter was used as the probe in this experiment?"
(4) Provide a brief comparison of WRKY28's role in B. napus versus other species to contextualize its functional conservation or divergence in “introduction.”
(5) minor comments
line 22: Replace "shooting branching" with "shoot branching."
line 204: Replace the phrase 'completely weakened' with a more precise expression, as it is ambiguous.
Line 252: Replace "outgrowth" with "outgrowth.”
line 289: Replace "Fiigure 4A,B,C" with "Fiigure 5A,B,C.”
Author Response
Point 1: Many figures in the manuscript are too small, making it difficult to discern details. The following improvements are strongly recommended:
Figure 1
- Increase the font size of labels and axes for better readability, especially in the phylogenetic tree and sequence alignment sections.
- For the motif analysis (Figure 1C), provide an enlarged version to improve the visibility of the sequence motifs.
Response:We appreciate the reviewer's suggestions. Figure 1 consists of four parts that collectively exhibits the sequence information of five copies of BRC1 in rapeseed. Inserting the combined Figure 1 into the text results in smaller details in the figure under the normal reading display ratio. We have uploaded a separate, original combination of Figure 1. Could the reviewer please blow up the reading display ratio or review the original image. If it still does not meet the requirements, we will modify it according to the requirements.
Figure 3
- In Figure 3A, enlarge the schematic representation of promoter sequences and adjust the text and annotations for better readability.
- The EMSA band shift diagram (Figure 3B) should be enlarged, with clearer labels and a more detailed legend.
- Enhance the resolution of fluorescence images in Figures 3C–H and ensure scale bars are clearly visible.
Response:Thank the reviewer for these comments. We have divided Figure 3 into two figures and made corresponding modifications of subsequent figure/image names in the revised manuscript. But in order to preserve the traces of modifications made to other suggestions, we have not completely removed the caption of Figure 3 in the original version. It should be deleted after this modification suggestion is adopted. In addition, We have added explanation for the shifted band in the figure legend of the revised manuscript.
Figure 4
- In Figure 4C-F, use different colors for the arrows to clearly distinguish between main stem buds and axillary buds.
Response:We are very grateful to the reviewer for valuable suggestions and insightful comments. The main stem buds and axillary buds are indicated by yellow arrows and magenta arrows, respectively, and have been modified accordingly in the revised manuscript and explained in the figure legend.
Point 2: Specify the number of biological replicates used in EMSA, luciferase assays, and RT-qPCR to enhance reproducibility.
Response:Thank you for the reviewer's excellent advice. Three biological replicates are used in EMSA, luciferase assays and RT-qPCR. And we have provided additional explanations in the “Materials and Methods” section in the revised manuscript.
Point 3: Figure 3A and B: The figure legend states that two types of promoters, BnaA01.BRC1 and BnaC01.BRC1, were analyzed. However, the figure seems to show results for only one. Which promoter was used as the probe in this experiment?"
Response:We apologize for the confusion caused to the reviewer. BnaA01.BRC1 and BnaC01.BRC1 are conserved in the promoter region containing the W-box, and the sequences used as probes are the same. We have provided explanations in the figure legend of the revised manuscript.
Point 4: Provide a brief comparison of WRKY28's role in B. napus versus other species to contextualize its functional conservation or divergence in “introduction”
Response:The gene homologous to BnaWRKY28 and well documented is AtWRKY28. We have listed reference 32 and 33 to introduce the representative role of AtWRKY28 including involve in the fate determination of megaspore mother cells and the regulation of leaf senescence.
Point 5: minor comments
line 22: Replace "shooting branching" with "shoot branching."
Response:Thank the reviewer for giving this suggestion. We have replaced "shooting branching" with "shoot branching" in the revised manuscript.
line 204: Replace the phrase 'completely weakened' with a more precise expression, as it is ambiguous.
Response:Thank the reviewer for pointing out this issue. We have replaced “completely weakened” with “diluted” in the revised manuscript.
Line 252: Replace "outgrowthed" with "outgrowth.”
Response:Thank the reviewer for pointing out this mistake. We have replaced “outgrowthed” with “outgrowth” in the revised manuscript.
line 289: Replace "Figure 4A,B,C" with "Figure 5A,B,C.”
Response:Thank the reviewer for pointing out this error. We have made correction in the revised manuscript.

Reviewer 2 Report
Comments and Suggestions for Authors
The paper investigates the involvement of WRKY28-BRC1 transcription factor complex in shoot branching in Brassica napus.
The paper's structure and experimental design are both clear, and the objectives of the authors are evident. However, minor revisions are required to enhance clarity and readability for the reader.
Line 111: it is supposed that the intended meaning should have been: “as a downstream player (or partner ?) of WRKY28”;
Line 124: typo? “on the apical meristem e clade”;
Figure 2A and S2. Nor in the text neither in the captions is clearly explained what does “Relative expression level” on Y axis of the graph means. Relative to which gene, or treatment, or line (overexpressing or knock out line);
Figure 3A. In the A explanation it is mentioned: “(A) Characterization of the promoter sequences of five BRC1 copies of rapeseed and the positions 221 of W-box…”. From the picture it is relatively clear that the characterization is for C03, A01, C01 but the other two are missing in the picture;
D, F, H it is not clear from the picture or from Material and Methods section if the “Relative luminescence intensity is derived from image analysis or luminometer;
I explanation is not consistent with the other figure subsection (e.g. (A), (B), (C));
In order to make clearer images and clearer captions I suggest dividing Figure 3 in two different figures: one for in vitro promoter analysis and one for in vivo promoter analysis;
Lines 283 and 284. Gene’s acronym not introduced (for the first time in the text) by the extended name of gene:
Figure 5. Caption missing (H) and (I);
Materials and Methods: description to obtain WIRKY28 and knock-out mutant lines missing;
Line 390: space between “with” and “1000”;
Line 440: typo for the word “fixed”;
Abbreviations: acronym for the cited genes should be added.
Author Response
Point 1: Line 111: it is supposed that the intended meaning should have been: “as a downstream player (or partner ?) of WRKY28”;
Response:Thank the reviewer for the excellent comment and suggestion. We have modified it to “as a downstream player of WRKY28” in the revised manuscript.
Point 2: Line 124: typo? “on the apical meristem e clade”;
Response:We respect to the reviewer for the preciseness and apologize for our carelessness. It should be “on the same clade”, and we have made corrections in the revised manuscript.
Point 3: Figure 2A and S2. Nor in the text neither in the captions is clearly explained what does “Relative expression level” on Y axis of the graph means. Relative to which gene, or treatment, or line (overexpressing or knock out line);
Response:Thank the reviewer for the rigorous examination. We did forget to specify the reference for calculating the relative expression levels, which is necessary to understand the meaning of the figures. In Figure 2A, the mean values of BnaA03.BRC1 were used to normalize the expression levels; and in Figure S2, the mean values of BnaA01.BRC1 in leaf were used to normalize the expression levels. The above explanations have been added in the captions in the revised manuscript.
Point 4: Figure 3A. In the A explanation it is mentioned: “(A) Characterization of the promoter sequences of five BRC1 copies of rapeseed and the positions of W-box…”. From the picture it is relatively clear that the characterization is for C03, A01, C01 but the other two are missing in the picture;
Response:We are very grateful to reviewer for the insightful comment. We analyzed the 2,000 bp sequences upstream of start codon ATG of the five rapeseed BRC1 copies. The BnaA01.BRC1 and BnaC01.BRC1 copies were sequence conserved, both containing a typical W-box at approximately -260 bp. The BnaC03.BRC1 copy contained two typical W-boxes at approximately -1040 bp and -1115 bp. However, no W-boxes were found in the BnaA03.BRC1 and BnaC05.BRC1 copies. Actually, Figure A showed a schematic diagram of the W-boxes distribution in BnaA01.BRC1, BnaC01.BRC1 and BnaC03.BRC1 copies. We have corrected “five BRC1 copies” to “three BRC1 copies” in the caption in the revised manuscript.
Point 5: D, F, H it is not clear from the picture or from Material and Methods section if the “Relative luminescence intensity is derived from image analysis or luminometer;
Response:We thank the reviewer for this comment. We explained in the caption that the data D, F and H were collected the fluorescence values from the leaves of C, E and G by InDiGo software, respectively; that is, relative luminescence intensity is derived from image analysis. However, word “leaves” cause confusion and ambiguity. We have changed “leaves” to “images” in the caption in the revised manuscript.
Point 6: I explanation is not consistent with the other figure subsection (e.g. (A), (B), (C));
Response:Thank the reviewer for pointing out this issue. We have corrected it.
Point 7: In order to make clearer images and clearer captions I suggest dividing Figure 3 in two different figures: one for in vitro promoter analysis and one for in vivo promoter analysis;
Response:Thank the reviewer for the valuable advice and these points are accepted. We have divided Figure 3 into two figures and made corresponding modifications of subsequent figure/image names in the revised manuscript. But in order to preserve the traces of modifications made to other suggestions, we have not completely removed the caption of Figure 3 in the original version. It should be deleted after this modification suggestion is adopted.
Point 8: Lines 283 and 284. Gene’s acronym not introduced (for the first time in the text) by the extended name of gene:
Response:Thank the reviewer for this comment. In the “Introduction”, when the LAS, RAX and CUC genes are first mentioned, the extended name of gene are introduced in line 43, 44 and 47. However, we did not list these genes in “Abbreviations”. We have added the acronyms for these genes in the revised manuscript.
Point 9: Figure 5. Caption missing (H) and (I);
Response:Thank the reviewer for pointing out this issue. We made writing error in the caption, with the wrong numbering. Two E and F appeared, and the second E and F should be H and I, respectively. We have made corrections in the revised manuscript.
Point 10: Materials and Methods: description to obtain WRKY28 and knock-out mutant lines missing;
Response:Thank the reviewer for pointing out this issue. We have added the description of obtaining WRKY28 transgenic line of rapeseed in “Materials and Methods” in the revised manuscript.
Point 11: Line 390: space between “with” and “1000”;
Response:Thank the reviewer for pointing out this reviewer for. We have made corrections in the revised manuscript.
Point 12: Line 440: typo for the word “fixed”;
Response:Thank the reviewer for pointing out this reviewer for. We have made corrections in the revised manuscript.
Point 13: Abbreviations: acronym for the cited genes should be added.
Response:Thank the reviewer for this suggestion. We have added the acronyms for the cited genes in the revised manuscript.
